# Organic Salts Based on Isoniazid Drug: Synthesis, Bioavailability and Cytotoxicity Studies

**DOI:** 10.3390/pharmaceutics12100952

**Published:** 2020-10-10

**Authors:** Filipa Santos, Luís C. Branco, Ana Rita C. Duarte

**Affiliations:** LAQV, REQUIMTE, Departamento de Química da Faculdade de Ciências e Tecnologia, Universidade Nova de Lisboa, 2829-516 Caparica, Portugal; mfilipacsantos@hotmail.com (F.S.); aduarte@fct.unl.pt (A.R.C.D.)

**Keywords:** ionic liquids & organic salts, tuberculosis, isoniazid, bioavailability, cytotoxicity

## Abstract

Tuberculosis is one of the ten causes of morbidity and mortality worldwide caused by *Mycobacterium tuberculosis* complex. Some of the anti-tuberculosis drugs used in clinic studies, despite being effective for the treatment of tuberculosis, present serious adverse effects as well as poor bioavailability, stability, and drug-resistance problems. Thus, it is important to develop approaches that could provide shorter drug regimens, preventing drug resistance, toxicity of the antibiotics, and improve their bioavailability. Herein, we reported the use of organic salts based on the isoniazid drug, which can act as an organic cation combined with suitable organic anions such as alkylsulfonate-based (mesylate, R or S-Camphorsulfonate), carboxylate-based (glycolate, vanylate) and sacharinate. The synthesis, characterization, and cytotoxicity studies comparing with the original isoniazid drug have been performed. The possibility to explore dicationic salts seems promising in order to improve original bioavailability, and promote the elimination of polymorphic forms as well as higher stability, which are relevant characteristics that the pharmaceutical industry pursues.

## 1. Introduction

The pharmaceutical industry and research community are continuously trying to improve their approaches and redesign drugs to enhance their therapeutic efficiency, while at the same time reducing the waste produced in the research and manufacture of drugs. The use of alternative solvents is one of the approaches that has been explored over the years for pharmaceutical and medicinal applications, namely in the improvement and modulation of the pharmacokinetics of drugs. Ionic systems such as ionic liquids and organic salts as well as eutectic mixtures are examples of alternative solvents that have been commonly investigated in pharmaceutical research [1]. Several studies reported that these solvents can avoid polymorphic forms of the drugs, and improve the solubility, permeability, and therapeutic activity of different active pharmaceutical ingredients (APIs) [2,3,4,5,6,7,8]. Lately, the use of organic salts and ionic liquids (OSILs) for therapeutic applications has been explored, using an API in its ionic form (e.g., ampicillin, [9,10,11] zoledronic acid [12], and ibuprofen [13]) and combining it with biocompatible counterions, producing API-OSILs. The use of API-OSILs allows the design of improved drugs that may modify the functionality and can provide important characteristics to the original API [6,14,15]. Usually, these API-OSILs present higher stability, solubility, dissolution rate and permeability when compared to the pure drug itself, however counterions added to the API could confer some potential toxicity that may lead to dose adjustments and more specific tests for the safety evaluation of these API-OSILs should be required [6].

The tuberculosis infection caused by *Mycobacterium tuberculosis* complex was first discovered by Robert Koch in 1882 and it is still considered by the World Health Organization (WHO) one of the infections that causes higher mortality rates worldwide by a single infectious agent [16,17,18,19]. In 2018, the WHO estimated 10 million new infections by tuberculosis in the world and 1.5 million deaths from tuberculosis infection [18]. The treatment of this infection includes a prolonged regimen of administration of several antibiotics and the challenges that tuberculosis treatment still faces are related to the decrease of treatment time, improvement of the stability of the drugs, and finding alternatives that could reduce cumulative adverse effects and prevent multidrug resistance. The patient adherence to this prolonged treatment is one of the barriers that affects the efficacy of the tuberculosis therapy, which makes it crucial to find forms that could reduce the time of the treatment and be more patient compliant, i.e., well tolerated treatments [19,20].

Drug resistance to an antibiotic result in the inefficacy of the antibiotic for treating the infection, since the bacteria do not respond to one or more antibiotics and frequently suffer from genetic changes and “re-adapt” their response when exposed to these drugs, will comprise a higher risk for spreading an infection and make the infection more difficult to treat. In case of tuberculosis infection, the development of drug resistance is mostly due to genetic modifications in genes that encode drug targets, impermeability of their cell wall, and activity of efflux pumps [21,22]. Multidrug resistance to isoniazid and rifampicin frequently occurs when they are used for first-line treatment of tuberculosis and are considered effective in killing the bacteria and controlling the infection. The multidrug resistance (MDR) to first-line anti-tuberculosis drugs leads to use of other antibiotics to treat the tuberculosis infection that comprises more adverse effects and more expensive, complex, and longer treatments for controlling the infection [23]. In addition, in tuberculosis therapy, an extensive drug resistance (XDR) may develop that is characterized by MDR to isoniazid, rifampicin, second-line anti-tuberculosis drugs, and fluoroquinolones, which restricts the therapeutic options for an effective treatment of the tuberculosis infection [21] and makes it imperative to find alternatives and solutions for an effective treatment of tuberculosis.

In tuberculosis therapy, different approaches have been reported to modulate the properties of some anti-tuberculosis drugs. Moreover, studies have been made with a first-line anti-tuberculosis drug, isoniazid, a prodrug used for prevention and treatment of tuberculosis infection since the 1950s [23]. This drug presents high resistance is associated with mutations in its activator enzyme (KatG, a multifunctional catalase–peroxidase that activates isoniazid and leads to the inhibition of inhA (enoyl-acyl-carrier protein (ACP) reductase) and consequently, to the inhibition of the biosynthesis of mycolic acids) [23,24]. Machado and coworkers mentioned that besides mutations that occur in *inhA* gene and cause resistance to the isoniazid drug, another genes could suffer from mutations and contribute to isoniazid resistance, like *ndh*, *kasA* and *oxyR-ahpC* from the intergenic region [22,25]. In spite of the low stability and fast degradation when administered with other anti-tuberculosis drugs, [26,27,28] isoniazid continues to be an essential drug recommended by WHO to integrate different drug regimens to treat tuberculosis infections [23]. Some studies made with this drug showed an improvement of the isoniazid solubility by simple dissolution of API in different imidazolium-based ionic liquids, observing that the solubility of the isoniazid decreases with the increase of the alkyl chain in the ionic liquid cation [29]. Another approach reported in the literature to improve the physicochemical properties and stability of isoniazid is the use of cocrystals with isoniazid, combining with numerous conformers like *p*-hydroxybenzoic acid, [30] nicotinamide, [30] fumaric acid, [30] succinic acid, [30] vanillic acid, [27] ferulic acid, [27] caffeic acid [27], and resorcinol [27]. However, to our knowledge, it was not reported in any study that included isoniazid in the ionic liquid structure by protonation approach or combination with a biocompatible counterion forming API-OSILs.

Considering the importance of contributing to the enhancement of tuberculosis therapy for better adherence of the patients to the treatment, finding well-tolerated drugs as well as more stable formulations is mandatory. The main goal of this work is the synthesis and characterization of different API-OSILs based on isoniazid cation and the subsequent study of their bioavailability (solubility and permeability studies) and cytotoxicity profiles. The most promising API-OSILs can be interest for further studies using *Mycobacterium tuberculosis*.

## 2. Materials and Methods

### 2.1. Materials

Isonicotinic acid hydrazide (CAS no. 54-85-3, 98% purity) was purchased from Alfa Aesar (Kandel, Germany) and was used as a cation for combination with the organic anions. The reagents used were commercially available and were purchased from Alfa Aesar and Sigma Aldrich (St. Louis, MO, USA). The organic salts prepared in this work were isoniazid chloride [INH][Cl] in the ratio 1:1 and 1:2, isoniazid mesylate [INH][MsO], isoniazid glycolate [INH][GcO], isoniazid camphorsulfonate S and R [INH][S-CsO] and [INH][R-CsO] in the ratio 1:1 and 1:2, isoniazid vanillate [INH][VanO], and isoniazid saccharinate [INH][Sac].

Phosphate buffered saline (PBS) solution was prepared from PBS tablets (Fisher BioReagents, Fair Lawn, NJ, USA) as indicated: one tablet dissolved in 200 mL of deionized water, yielding 0.01 M phosphate buffer, 0.0027 M potassium chloride, 0.137 M sodium chloride, pH 7.4 solution at 25 °C.

### 2.2. Preparation of Organic Salts Based on Isoniazid

#### 2.2.1. Isoniazid Monochloride [INH][Cl]

Isoniazid (1.0682 g; 7.79 mmol; 1 eq.) was dissolved in 20 mL of deionized water, then a solution of HCl 1M (7.8 mL; 7.79 mmol; 1 eq.) was slowly added to the isoniazid solution. The reaction mixture was stirred at room temperature for 3 h. Then, the solvent was evaporated, and the final product was dried under vacuum. The product was obtained as a pale pink solid (1.235 g; 91.3%). ^1^H NMR (400.13 MHz, DMSO-_d6_): δ = 8.83 (d, 2H, J = 4.0 Hz), 7.90 (dd, 2H, J = 4.0 Hz), (Appendix A). ^13^C NMR (100.62 MHz, DMSO-_d6_): δ = 164.39, 150.32, 140.67, 138.88, 122.32, 122.26 ppm (Appendix A). IR: 3099, 2951, 1981, 1658, 1533, 1493, 1316, 1246, 1141, 1063, 1012, 967, 863, 758, 690, 660, 607, 489, 409 cm^−1^ (Appendix A). Anal. calcd. for C_6_H_8_ClN_3_O C 41.51, H 4.65, N 24.21, found: C 41.48, H 4.60, N 24.66. M.p. 93 °C.

#### 2.2.2. Isoniazid Dichloride [INH][Cl]_2_

Isoniazid (1.1477 g; 8.37 mmol; 1 eq.) was dissolved in 20 mL of deionized water, then a solution of HCl 1M (16.7 mL; 16.74 mmol; 2 eq.) was slowly added to the isoniazid solution. The reaction mixture was stirred at room temperature for 3 h. Then, the solvent was evaporated, and the final product was dried under vacuum. The product was obtained as pale white solid (1.1438 g; 64.8%). ^1^H NMR (400.13 MHz, DMSO-_d6_): δ = 8.92 (d, 2H, J = 4.0 Hz), 8.06 (d, 2H, J = 4.0 Hz), (Appendix A). ^13^C NMR (100.62 MHz, DMSO-_d6_): δ = 164.34, 150.67, 140.61, 140.56, 121.79, 121.48 ppm (Appendix A). IR: 3094, 3050, 2950, 2500, 2074, 1991, 1684, 1637, 1609, 1565, 1515, 1496, 1360, 1311, 1243, 1211, 1184, 1156, 1100, 1059, 1039, 1009, 869, 835, 750, 682, 561, 505, 419 cm^−1^ (Appendix A). Anal. calcd. for C_6_H_9_Cl_2_N_3_O C 34.31, H 4.32, N 20.00, found: C 34.36, H 4.40, N 19.85. M.p. 160 °C.

#### 2.2.3. Isoniazid Mesylate [INH][MsO]

Isoniazid (1.034 g; 7.54 mmol; 1 eq.) was dissolved in 20 mL of deionized water, then a solution of mesylic acid 1M (7.5 mL; 7.54 mmol; 1 eq.) was slowly added to the isoniazid solution. The reaction mixture was stirred at room temperature for 3 h. Then, the solvent was evaporated, and the final product was dried under vacuum. The product was obtained as pale white solid (1.7484 g; 99.5%). ^1^H NMR (400.13 MHz, DMSO-_d6_): δ = 11.24 (s, 1H), 8.96 (d, 2H, J = 8.0 Hz), 8.09–8.06 (m, 2H), 2.44–2.42 (m, 3H), (Appendix A). ^13^C NMR (100.62 MHz, DMSO-_d6_): δ = 163.83, 148.42, 142.61, 124.73, 123.27, 53.34 ppm (Appendix A). IR: 3424, 3191, 2985, 1676, 1654, 1540, 1497, 1419, 1298, 1223, 1152, 1061, 1030, 1001, 934, 846, 777, 744, 686, 664, 551, 535, 522, 459, 426, 403 cm^−1^ (Appendix A). Anal. calcd. for C_7_H_11_N_3_O_4_S C 36.05, H 4.75, N 18.02 found: C 36.83, H 4.90, N 18.05. M.p. 133 °C.

#### 2.2.4. Isoniazid Glycolate [INH][GcO]

Isoniazid (1.2722 g; 9.28 mmol; 1 eq.) was dissolved in 20 mL of deionized water, then the glycolic acid was weighed (0.705 g; 9.28 mmol; 1 eq.) and slowly added to isoniazid solution. The reaction mixture was stirred at room temperature for 3 h. Then, the solvent was evaporated, and the final product was dried under vacuum. The product was obtained as white solid (1.8670 g; 94%). ^1^H NMR (400.13 MHz, DMSO-_d6_): δ = 10.10 (s, 1H), 8.71 (d, 2H, J = 4.0 Hz), 7.74 (d, 2H, J = 8.0 Hz), 3.90 (s, 2H), (Appendix A). ^13^C NMR (100.62 MHz, DMSO-_d6_): δ = 171.49, 164.30, 150.86, 139.99, 139.75, 121.78, 121.45, 61.22 ppm (Appendix A). IR: 3370, 3249, 1652, 1605, 1539, 1511, 1495, 1417, 1343, 1304, 1226, 1080, 1061, 1001, 983, 952, 847, 744, 685, 663, 577, 513, 436 cm^−1^ (Appendix A). Anal. calcd. for C_8_H_11_N_3_O_4_ C 45.07, H 5.20, N 19.71 found: C 46.68, H 4.81, N 20.16.

#### 2.2.5. Isoniazid mono(S-Camphorsulfonate) [INH][S-CsO]

Isoniazid (1.1098 g; 8.09 mmol; 1 eq.) was dissolved in 20 mL of deionized water, then the S-camphor sulfonic acid was weighed (1.88 g; 8.09 mmol; 1 eq.) and slowly added to the isoniazid solution. The reaction mixture was stirred at room temperature for 3 h. Then, the solvent was evaporated, and the final product was dried under vacuum. The product was obtained as yellow solid (2.8815 g; 96.3%). ^1^H NMR (400.13 MHz, DMSO-_d6_): δ = 11.10 (s, 1H), 8.96–8.86 (m, 2H), 8.41–8.15 (m, 2H), 3.47–3.22 (m, 1H), 2.78–2.65 (m, 2H), 2.45–2.39 (m, 1H), 2.33 (d, 1H, J = 16.0 Hz), 2.04–1.79 (m, 2H), 1.64–1.57 (m, 1H), 1.30–1.23 (m, 1H), 0.98 (t, 3H, J = 20.0, 32.0 Hz), 0.75 (t, 3H, J = 20.0, 32.0 Hz), (Appendix A). ^13^C NMR (100.62 MHz, DMSO-_d6_): δ = 178.51, 162.13, 146.06, 143.63, 127.36, 124.35, 55.05, 50.61, 49.41, 43.80, 43.23, 28.53, 26.91, 19.90, 19.67 ppm (Appendix A). IR: 3429, 2584, 1674, 1651, 1559, 1501, 1304, 1259, 1240, 1193, 1150, 1101, 1022, 995, 930, 842, 792, 764, 664, 647, 609, 590, 539, 519, 451 cm^−1^ (Appendix A). Anal. calcd. for C_16_H_23_N_3_O_5_S.H_2_O C 49.60, H 6.50, N 10.85 found: C 50.83, H 6.32, N 11.09.

#### 2.2.6. Isoniazid di(S-Camphorsulfonate) [INH][S-CsO]_2_

Isoniazid (0.7752 g; 5.65 mmol; 1 eq.) was dissolved in 20 mL of deionized water, then the S-camphor sulfonic acid was weighed (2.83 g; 11.3 mmol; 2 eq.) and slowly added to the isoniazid solution. The reaction mixture was stirred at room temperature for 3 h. Then, the solvent was evaporated, and the final product was dried under vacuum. The product was obtained as yellow solid (3.0096 g; 88.5%). ^1^H NMR (400.13 MHz, DMSO-_d6_): δ = 11.17 (s, 1H), 9.04–9.00 (m, 2H), 8.32–8.30 (m, 2H), 3.26–2.87 (m, 2H), 2.79–2.72 (m, 2H), 2.49–2.48 (m, 4H), 2.41–2.23 (m, 2H), 2.05–1.95 (m, 2H), 1.90–1.67 (m, 2H), 1.35–1.29 (m, 2H), 1.04–0.97 (m, 6H), 0.83–0.67 (m, 6H), (Appendix A). ^13^C NMR (100.62 MHz, DMSO-_d6_): δ = 216.41, 216.38, 191.45, 149.19, 144.27, 142.61, 125.35, 56.33, 55.23, 51.40, 50.96, 49.57, 49.06, 47.61, 47.38, 43.16, 43.11, 42.69, 42.60, 26.84, 24.65, 20.43, 19.97, 19.48, 19.40 ppm (Appendix A). IR: 3110, 1635, 1590, 1531, 1412, 1293, 1248, 1149, 1030, 970, 831, 742, 668, 598, 524 cm^−1^ (Appendix A). Anal. calcd. for C_26_H_39_N_3_O_9_S_2_ C 51.90, H 6.53, N 6.98 found: C 51.57, H 6.26, N 7.48.

#### 2.2.7. Isoniazid mono(R-Camphorsulfonate) [INH][R-CsO]

Isoniazid (1.2520 g; 9.13 mmol; 1 eq.) was dissolved in 20 mL of deionized water, then the R-camphor sulfonic acid was weighed (2.12 g; 9.13 mmol; 1 eq.) and slowly added to the isoniazid solution. The reaction mixture was stirred at room temperature for 3 h. Then, the solvent was evaporated, and the final product was dried under vacuum. The product was obtained as pale white solid (3.2321 g; 96.2%). ^1^H NMR (400.13 MHz, DMSO-_d6_): δ = 11.16 (d, 1H, J = 20.0 Hz), 8.98–8.87 (m, 2H), 8.41–8.00 (td, 2H, J = 4.0 Hz), 3.34–3.23 (dd, 1H, J = 12.0 Hz), 2.94–2.64 (m, 2H), 2.46–2.39 (m, 1H), 2.34–2.22 (m, 1H), 2.04–1,79 (m, 2H), 1.65–1.47 (m, 1H), 1.30–1.17 (m, 1H), 0.98 (t, 3H, J = 20.0, 32.0 Hz), 0.75 (t, 3H, J = 20.0, 32.0 Hz), (Appendix A). ^13^C NMR (100.62 MHz, DMSO-_d6_): δ = 178.45, 162.05, 145.60, 143.41, 127.43, 124.52, 55.02, 50.61, 49.40, 43.79, 43.22, 28.49, 26.90, 19.89, 19.64 ppm (Appendix A). IR: 3433, 1666, 1626, 1555, 1499, 1242, 1200, 1151, 1022, 1000, 838, 748, 677, 591, 537, 509, 405 cm^−1^ (Appendix A). Anal. calcd. for C_16_H_23_N_3_O_5_S.0.5H_2_O C 50.79, H 6.35, N 11.11 found: C 50.35, H 6.39, N 11.09.

#### 2.2.8. Isoniazid di(R-Camphorsulfonate) [INH][R-CsO]_2_

Isoniazid (1.0727 g; 7.82 mmol; 1 eq.) was dissolved in 20 mL of deionized water, then the R-camphor sulfonic acid was weighed (3.6369 g; 15.6 mmol; 2 eq.) and slowly added to the isoniazid solution. The reaction mixture was stirred at room temperature for 3 h. Then, the solvent was evaporated, and the final product was dried under vacuum. The product was obtained as yellow solid (4.164 g; 88.7%). ^1^H NMR (400.13 MHz, DMSO-_d6_): δ = 11.17 (s, 1H), 9.04–8.92 (m, 2H), 8.47–8.15 (td, 2H, J = 4.0, 8.0 Hz), 3.26–3.21 (dd, 2H, J = 4.0 Hz), 2.96 (d, 2H, J = 16.0 Hz), 2.87–2.64 (m, 4H), 2.48–2.46 (m, 4H), 2.42–2.22 (m, 2H), 2.09–1.99 (m, 2H), 1.91–1.62 (m, 2H), 1.36–1.26 (m, 2H), 1.05–0.97 (m, 6H), 0.83–0.67 (m, 6H), (Appendix A). ^13^C NMR (100.62 MHz, DMSO-_d6_): δ = 216.51, 191.52, 147.16, 144.71, 142.75, 125.29, 124.97, 56.35, 55.25, 51.38, 51.02, 49.55, 49.01, 47.59, 47.30, 43.14, 43.12, 42.69, 42.59, 26.85, 24.62, 20.51, 19.99, 19.55, 19.44 ppm (Appendix A). IR: 3207, 3112, 2974, 1635, 1536, 1412, 1293, 1248, 1134, 1035, 841, 747, 668, 603, 524 cm^−1^ (Appendix A). Anal. calcd. for C_26_H_39_N_3_O_9_S_2_ C 51.90, H 6.53, N 6.98 found: C 52.56, H 6.48, N 7.54. M.p. 133 °C.

#### 2.2.9. Isoniazid Vanillate [INH][VanO]

Isoniazid (1.6023 g; 11.7 mmol; 1 eq.) was dissolved in 20 mL of deionized water, then the vanillic acid was weighed (1.964 g; 11.7 mmol; 1 eq.) and slowly added to the isoniazid solution. The reaction mixture was stirred at room temperature for 3 h. Then, the solvent was evaporated, and the final product was dried under vacuum. The product was obtained as pale brown solid (3.4564 g; 96.8%). ^1^H NMR (400.13 MHz, DMSO-_d6_): δ = 9.84 (s, 1H), 8.72 (d, 2H, J = 4.0 Hz), 7.74–7.73 (dd, 2H, J = 4.0 Hz), 7.46–7.44 (dd, 2H, 8.0 Hz), 6.86 (d, 1H, J = 8.0 Hz), 3.81 (s, 3H), (Appendix A). ^13^C NMR (100.62 MHz, DMSO-_d6_): δ = 167.66, 164.33, 151.54, 150.66, 147.66, 140.72, 123.92, 122.05, 121.45, 115.47, 113.16, 55.99 ppm (Appendix A). IR: 3244, 1674, 1591, 1512, 1484, 1451, 1408, 1330, 1288, 1268, 1235, 1202, 1170, 1122, 1064, 1022, 997, 888, 844, 780, 763, 750, 680, 581, 535, 471, 414, 404 cm^−1^ (Appendix A). Anal. calcd. for C_14_H_15_N_3_O_5_ C 55.08, H 4.95, N 13.76 found: C 55.38, H 4.91, N 13.74.

#### 2.2.10. Isoniazid Saccharinate [INH][Sac]

[INH][Cl] (1.0258 g; 5.9 mmol; 1 eq.) was dissolved in 20 mL of methanol, then the saccharin sodium salt was weighed (1.212 g; 5.9 mmol; 1 eq.) and slowly added to the [INH][Cl] solution. The reaction mixture was stirred at room temperature for 3 h. Then, the solvent was evaporated and precipitated the NaCl with acetone, the final product was dried under vacuum. The product was obtained as pale orange solid (1.7822 g; 94.3%). ^1^H NMR (400.13 MHz, DMSO-_d6_): δ = 11.19–10.80 (dd, 2H, J = 4.0 Hz), 8.83–8.78 (m, 2H), 8.01 (d, 1H, J = 8.0 Hz), 7.90–7.76 (m, 3H), 7.25 (s, 2H), (Appendix A). ^13^C NMR (100.62 MHz, DMSO-_d6_): δ = 167.89, 164.79, 150.77, 141.90, 139.74, 133.01, 132.65, 131.37, 130.17, 127.82, 121.99, 121.97 ppm (Appendix A). IR: 3192, 3112, 3008, 1625, 1536, 1412, 1327, 1288, 1248, 1134, 1124, 970, 841, 742, 663, 608, 534 cm^−1^ (Appendix A). Anal. calcd. for C_13_H_12_N_4_O_4_S.0.5H_2_O C 47.42, H 3.95, N 17.02 found: C 47.26, H 3.79, N 16.69. ICP for Na^+^ < 200 ppm. M.p. 179 °C.

### 2.3. Physical-Chemical Characterization

The ^1^H and ^13^C NMR were prepared in deuterated dimethyl sulfoxide (DMSO-_d6_ from Euriso-Top, St. Aubin Cedex, France) and were obtained on a Bruker Avance III 400 spectrometer (Bruker, Billerica, MA, USA) at an operating frequency of 400.13 MHz for ^1^H and 100.62 MHz for ^13^C and chemical shifts were referenced to Me_4_Si (δ in ppm). Infrared spectroscopy was recorded on a PerkinElmer spectrum Two (PerkinElmer, Waltham, MA, USA) in the range of 400–4000 cm^−1^. The thermograms were obtained through differential scanning calorimetry by a DSC Q200 (TA instruments) (New Castle, DE, USA) in aluminum hermetic pans with a pinhole. The measurements were made under dry nitrogen atmosphere (flow rate of 50 mL min^−1^) with samples between 3–10 mg. The samples were equilibrated at 25 °C and then 2 cycles for isoniazid and 3 cycles for organic salts in a range of temperatures between −90 °C and 180 °C were performed. Each cycle was performed with a ramp up to 180 °C with a heating rate of 10 °C min^−1^, followed by an isothermal period of 1 min, and cooling ramp to −90 °C with a cooling rate of 10 °C min^−1^. The quantity of sodium in samples of [INH][Sac] was measured by inductively coupled plasma-atomic emission spectrometer (ICP-AES) from Horiba Jobin-Yvon, Palaiseau, France, Ultima, a model equipped with a 40.68 MHz RF. The samples were characterized by elemental analysis too with 2–3 mg of sample with an Elemental analyzer Thermo Finnigan-CE Instruments Flash EA 1112 CHNS series (Italy).

### 2.4. Solubility Studies

The solubility of organic salts with isoniazid was performed in water and PBS at 37 °C. Briefly, an excess of organic salt was added to water and the PBS solution and stirred (60 rpm) for 24 h. The determination of solubility of organic salts was made by UV spectroscopy in a microplate reader (VICTOR Nivo ^TM^, PerkinElmer, Waltham, MA, USA). The absorbance of the solutions was measured at the maximum absorption wavelength of isoniazid (263 nm). The calibration curve was made using isoniazid as standard in each solvent for quantification (R^2^ = 0.9974 for water and R^2^ = 0.9993 for PBS) and all samples were measured in triplicates. The pH of aqueous solutions was measured using an analogic pH meter (914 pH/Conductometer, model 2.914.0220, Metrohm, Herisau, Switzerland) with a glass electrode coupled to a temperature sensor (NTC) (6.0228.010, Metrohm, Herisau, Switzerland).

### 2.5. Permeability Studies

The permeability studies were performed with Franz diffusion cells (PermeGear, Hellertown, PA, USA) with 8 mL in the receptor compartment and 2 mL in the donor compartment with an effective mass transfer area of 1 cm^2^. The membranes used were synthetic membranes, polyethersulfone (PES-U) membrane with 150 μm thickness and 0.45 μm pore size (Sartorius Stedim Biotech, Goettingen, Germany), that were placed between the two compartments and seized with a stainless steel clamp. The receptor compartment was filled with PBS solution and air bubbles were removed by carefully tilting the Franz cell, then the donor compartment was filled with the organic salt and 2 mL of PBS solution. Aliquots of 200 μL were taken from the receptor compartment in intervals of 15, 30, 45, 60, 75, 90, 105, 120, 150, 180, and 240 min, and fresh PBS was added to complete the volume. The experiments were performed at 37 °C in a water bath with stirring at 60 rpm, to eliminate the boundary layer effect. The permeability of the organic salts with isoniazid was measured by UV spectroscopy in a microplate reader, similarly to what was described for the solubility assay. The cumulative mass of the isoniazid diffused to receptor compartment was determined taking in consideration the addition of fresh PBS solution. The permeability (*P*) of the isoniazid and organic salts with isoniazid was calculated following Equation (1):(1)$$-\ln\left(1-\frac{2C_{t}}{C_{0}}\right)=\frac{2A}{V}\;\times \;P\;\times \;t$$
where *C_t_* is the concentration in the receptor compartment at time *t*, *C_0_* is the initial concentration in the donor compartment, *V* is the volume in two compartments, and *A* is the effective area of permeation. The permeability coefficient can be calculated from the slope of the curve *–(V/2A)*ln(1 − 2C_t_/C*_0_*)* versus *t* (cm s^−1^) [31,32,33,34].

The diffusion coefficient (*D*) of the isoniazid and organic salts across the membrane was calculated according to Fick’s law of diffusion, following the Equation (2):(2)$$D=\frac{V_{1}V_{2}}{V_{1}+V_{2}}\times \frac{h}{A}\times \frac{1}{t}\ln\left(\frac{C_{f}-C_{i}}{C_{f}-C_{t}}\right)$$
where *D* is the diffusion coefficient (cm^2^ s^−1^), *C_i_* and *C_f_* are the initial and final concentrations, and *C_t_* is the concentration at time *t* in the receptor compartment (mol L^−1^), *V_1_* and *V_2_* are the volume of liquid in donor and receptor compartment, respectively, *h* is the thickness of the membrane and *A* is the effective diffusion area of the membrane [31,32,33,34].

### 2.6. Cell Culture and Cytotoxicity Assays

To evaluate the biological performance of isoniazid and organic salts of isoniazid studies were carried out using both L929 and A549 cell lines. The L929 cells are mouse fibroblasts of connective tissue (DSMZ, ACC 2, Braunschweig, Germany) and A549 cells are human alveolar epithelial cells from human lung carcinoma (DSMZ, ACC 107, Braunschweig, Germany). Both cell lines were cultured in MEM media (Corning, Manassas, VA, USA), supplemented with 10% fetal bovine serum (FBS, Corning, USA) and 1% antibiotic–antimycotic solution (Corning, USA). The cells were sub-cultured in a humidified atmosphere of 37 °C with 5% of CO_2_.

The cytotoxicity assay was performed in accordance with ISO/EN 10993 guidelines, in which the cells were seeded onto a 96-well plate (1 × 10^5^ cells/mL in L929 cells and 2 × 10^5^ cells/mL in A549 cells) and after 24 h of attachment, the medium were removed and the cells were incubated with 100 μL of extracts with 100 mM, 25 mM, and 1 mM concentration of organic salts and isoniazid, during 20–24 h. After the incubation time, the cells were washed with PBS and 100 μL of MTS (3-(4,5-dimethylthiazol-2-yl)-5-(3-carboxymethoxyphenyl)-2-(4-sulfophenyl)-2*H*-tetrazolium)) media were added at each well, in a dilution of 1:10. Then they were incubated for 3–4 h, at 37 °C with 5% of CO_2_ atmosphere. The amount of formazan product was measured in a microplate reader at 490 nm. The data were expressed in percentage of cell viability with the control and the experiments.

## 3. Results

### 3.1. Synthesis and Characterization of Isoniazid-Based Salts

In the synthesis of isoniazid based on organic salts and ionic liquids (API-OSILs), the isoniazid drug was used as organic cation and then different counterions were selected. Two possible synthetic approaches involving direct protonation by suitable acid addition or anion-exchange reactions in optimized conditions were performed. No additional purification steps are required in the case of the direct protonation approach while in the case of the anion-exchange approach, the inorganic salt should be removed by precipitation using appropriate organic solvent. All obtained API-OSILs were characterized by ^1^H and ^13^C NMR, FTIR, and elemental analysis in order to prove the desired organic chemical structure as well as final purities. Additionally, ICP-AES technique was used in the case of anion-exchange reaction in order to quantify Na content in the final compounds. Scheme 1 illustrates the synthetic approaches and type of anions for API-OSILs.

In some cases, it was possible to prepare isoniazid in the mono- and di-cationic form and then characterize by ^1^H-NMR. It is important to mention the observation of a clear chemical shift from aromatic protons of neutral isoniazid (7.73 and 8.70 ppm) compared to mono-protonated (7.89 and 8.82 pp, for [INH](Cl]) and di-protonated structures (8.05 and 8.91 ppm for [INH][Cl]_2_), as observed in Figure 1. In FTIR-ATR spectra, it is detected that a characteristic peak at 3300 cm^−1^ corresponds to a stretching of N-H in hydrazide group. This group in protonated forms is not observed due to the vibrational changes that occur when the protonation occurs. For all FTIR spectra, it is also possible to observe the peaks in the region of 3100–2900 cm^−1^ attributed to stretching vibrations of heteroatoms groups, and in protonated compounds the region that appears between 2100–1900 cm^−1^ that gets more intense in dicationic forma, in this form a new band appears at 2500 cm^−1^.

### 3.2. Thermal Characterization

The melting points (Tm), crystallization (Tc), and glass transition (Tg) temperatures were measured using differential scanning calorimetry and are summarized in Table 1. It is observed that the isoniazid monochloride ([INH][Cl]) possesses the lowest melting temperature and a higher reduction in its melting point compared to starting isoniazid. In general, the other organic salts showed melting points lower than isoniazid, except for [INH][Sac] that presented a melting point slightly above the 173 °C of isoniazid. The organic salts containing [GcO], [S-CsO], [S-CsO]_2_, [R-CsO], and [VanO] anions seem to become amorphous after the first cycle. This is an important observation because the crystallization of isoniazid is suppressed using these different organic anions, reducing or eliminating its polymorphism behavior.

### 3.3. Solubility and Permeability Studies

The solubility studies were carried out in water and PBS at 37 °C to simulate the physiological conditions and the results are represented in Table 2 and Figure 2.

The anions that are double protonated present higher values of solubility and the corresponding anions monoprotonated present a solubility profile similar to isoniazid, except for [INH][R-CsO]. The other organic salts did not present an improvement on the solubility of isoniazid in the selected media and in the most cases present values that are lower than the API itself. Regarding the measurement of the pH of all organic salts dissolved in water, a decrease in pH was observed, present in an acidic pH, that is, lower than the pH presented with isoniazid.

The permeability studies were only carried out with the organic salts that seemed more promising in terms of solubility characteristics such as [INH][Cl] in both ratios and [INH][CsO] in the R and S form and in both ratios, and the results are presented in Table 3.

The permeability assays were performed with synthetic membranes as a first screening of diffusion coefficients and cumulative mass release. It was observed that the organic salts tested presented a permeability below the one presented for isoniazid, despite the organic salt in dicationic form, chloride, that presented a value slightly below the isoniazid. In terms of diffusion coefficient in the media it is noticeable that the dicationic form chloride presents a high diffusion coefficient and the other organic salts present a diffusion coefficient value near to the isoniazid itself, with exception of chloride monoprotonated that presents a small value of diffusion coefficient. With the organic salts composed of isoniazid diprotonated and camphorsulfonates, the values of permeability and diffusion coefficients were too low to allow an accurate measure, possibly due to the higher size of the molecules that cannot permeate the membrane.

### 3.4. Biological Performance

Cell viability of isoniazid and organic salts was evaluated in two different cell lines, L929 and A549, and in different concentrations between a high concentration of 100 mM and a much lower concentration of 1 mM (Figure 3). It was observed that with a concentration of 100 mM all organic salts and ionic liquids prepared compromised the cell viability in both cell lines. At a concentration of 25 mM, the salts with camphorsulfonic acids and diprotonated salt with chloride presented a high toxicity in both cell lines. With a concentration of 1 mM, all compounds tested present a good cell viability and are considered non toxic.

## 4. Discussion

Isoniazid is a first-line anti-tuberculosis drug that presents a proven efficacy in tuberculosis treatment but presents some serious adverse effects like hepatotoxicity and a high level of multidrug resistance. Despite isoniazid in the pure state being stable for long periods of time, when administrated with other anti-tuberculosis drugs, it interacts and becomes unstable and suffers some degradation. Recently, it was reported that two polymorphs that are formed by hydrogen bonding in pyridine group (N-H…N) and another one in hydrazide groups (N-H…O) [35]. This anti-tuberculosis drug may be sensitive to light, when exposed for long periods of time, and is an active substance that, usually, is vulnerable to hydrolysis, oxidation, and interaction with other compounds (e.g., excipients used in formulation) to form hydrazones [27,36]. With the knowledge of the problems of stability and interaction with other drugs that isoniazid presents, it is important to explore new approaches that could overcome the problems of this effective anti-tuberculosis drug.

In this work, a new strategy for synthesis of organic salts using isoniazid drug as cation and combined with sulfonate and carboxylate counterions was explored. This new approach for the synthesis of API-OSILS with the isoniazid drug has the main goal of stabilizing this drug and avoiding polymorphic forms [35] and to understand if its properties could be tuned by the protonation of isoniazid with biocompatible counterions. The counterions selected to be combined with isoniazid were chosen as they are generally considered biocompatible and are commercially available [37,38]. The synthesis of the API-OSILS was performed either by protonation with acids or by metathesis reaction when salts were used for the preparation of API-OSILs. With ^1^H-NMR, FTIR-ATR, and elemental analysis it was possible to differentiate the compounds from monoprotonated to diprotonated and observe, for example in ^1^H-NMR, some deviations of the chemical shift that occurs in a more visible way in the diprotonated form of [INH][Cl]_2_. The thermal events were observed by DSC and the thermograms showed that the organic salts prepared have melting points below the melting point of isoniazid with the exception of [INH][Sac] that present a melting point of 179 °C that is slightly superior to isoniazid itself. However, with salts prepared with [GcO], [R-CsO], [S-CsO], [S-CsO]_2_, and [VanO]), it was observed that after the first cycle these compounds become amorphous, which could represent an advantage in the use of these compounds, since they avoid polymorphic forms and could make the drug more stable.

Another important property to measure with these new compounds is their solubility, as this characteristic is essential for pharmacological approaches, drug delivery, and for achieving an adequate response of the organism to the drug [39]. In this study, the solubility was measured as the solubility of these compounds in comparison with the isoniazid itself. It was observed that the double protonation forms present higher solubility in both solvents tested at 37 °C. The organic salts [INH][Cl], [INH][Cl]_2_, [INH][S-CsO], [INH][S-CsO]_2_, [INH][R-CsO] and [INH][R-CsO]_2_ appear to present a higher solubility in both solvents tested in comparison with other organic salts, which lead us to evaluate the permeability of these salts in synthetic membranes and determine their permeability over time and see if a higher solubility could lead to a higher driving force through the membrane. The permeability assays were performed with the Franz cells method for the first screening of the permeability through a synthetic membrane of the API and organic salts with isoniazid and the passive diffusion of the compounds through the membrane was measured [40]. The diffusion profiles present an initial lag phase in all compounds tested, due to the time needed for the solute to cross the membrane and start to diffuse in the media. In terms of permeability it was observed that only [INH][Cl]_2_ have a permeability that nears the value of isoniazid. The diffusion coefficient, which estimates the mass of compound that is diffused over time, is higher in the salt [INH][Cl]_2_ that doubles the diffusion when compared with the monoprotonated form, and presents to be higher than isoniazid itself, which is related to a faster diffusion through the membrane for this compound [33]. However, the other dicationic forms prepared with camphorsulfonates are not able to pass through the membrane, probably due to the higher size of the molecule that makes them unable to cross the synthetic membrane.

The cell viability assays performed show that at higher concentrations of isoniazid and organic salts (100 mM) were considered toxic for both cell lines. Despite camphorsulfonates and dicationic chloride salts presenting a good solubility of isoniazid, a high toxicity at 100 mM and 25 mM is observed. Nevertheless, when used in smaller concentrations these compounds present a good cell viability and are considered non toxic for both cell lines tested.

## 5. Conclusions

Tuberculosis treatment continues to be a challenge for biomedical research due to multidrug resistance of anti-tuberculosis drugs used as first and second-line treatment and several adverse effects and prolonged treatments that compromise the compliance of the treatment by patients with tuberculosis disease. Herein, the development of new compounds or the improvement of the bioavailability of existing drugs could be helpful for controlling multidrug resistance and decreasing the adverse effects of anti-tuberculosis drugs. In this study, the synthesis of organic salts using isoniazid as a cation seems to be a promising approach for the enhancement of some of the physical, thermal, and biological properties of this anti-tuberculosis drug, mainly when isoniazid is double protonated. From this work it can be concluded that dicationic forms can contribute to enhancing the characteristics of the drugs, such as solubility and stability, that turn these compounds with potential for future testing in *mycobacteria* strains envisaging tuberculosis therapy. Furthermore, the protonation of the API can further promote the elimination of polymorphic forms and higher stability of the drugs, which are some of the main characteristics that the pharmaceutical industry pursues.

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
