# Peer review of "Organic Salts Based on Isoniazid Drug: Synthesis, Bioavailability and Cytotoxicity Studies"

_pharmaceutics, 2020, doi:10.3390/pharmaceutics12100952_

Round 1

Reviewer 1 Report

Santos et al synthesized organic salts based on isoniazid drug combined with coordinating and noncoordinating anions based on alkylsulfonates and sugar-derivates. Authors have characterized the analogs well and tested them for solubility, cell permeability and cell viability.

Comments

  1. Add current statistics of tuberculosis.
  2. Include information on drug resistance or multidrug resistance in TB
  3. Talk about INH resistance and how these drugs will improve INH or INH based treatment
  4. Add physical characteristics such as melting point, UV absorbance
  5. I suggest authors perform hydrolytic stability at different pH of these analogs as IHN is known to get hydrolyzed.
  6. Figure 3: Calculate IC50 values. At 1 mM authors showed that compounds are not toxic. I was wondering if they produce antitubercular activity at that dose.

Author Response

Thank you for your comments.

Please consider the following detailed revision:

  1. Add current statistics of tuberculosis.

The authors recognize the importance of mention these data and reformulated the text and add the statistics of tuberculosis.

“In 2018, the WHO estimated 10 million of new infections by tuberculosis in the world and 1.5 million deaths from tuberculosis infection.”

  1. Include information on drug resistance or multidrug resistance in TB.

The authors acknowledge the comment and reformulated the text and include information about drug resistance in TB.

“Drug-resistance to an antibiotic result in the inefficacy of the antibiotic for treating the infection, since the bacteria do not respond to one or more antibiotics and, frequently suffer from genetic changes and “readapt” its response when exposed to these drugs, which will comprise a higher risk for spreading an infection and make the infection more difficult to treat. In case of tuberculosis infection, the development of drug-resistance is mostly due to genetic modifications in genes that encode for drug targets, impermeability of its cell wall and activity of efflux pumps. [21], [22] It is frequent occur multidrug-resistance to isoniazid and rifampicin, that are used for first-line treatment of tuberculosis and are considered effective in killing the bacteria and controlling the infection. The multidrug-resistance (MDR) to first-line anti-tuberculosis drugs lead to use of another antibiotics for treat the tuberculosis infection that comprises more adverse effects and more expensive, complex and longer treatments for controlling the infection.[23] In addition, in tuberculosis therapy, it may be developed an extensively drug-resistance (XDR) that is characterized by MDR to isoniazid, rifampicin, second-line anti-tuberculosis drugs and fluoroquinolones, which leaves restricted the therapeutic options for an effective treatment of the tuberculosis infection[21] and turns imperative find alternatives and solutions for an effective treatment of tuberculosis.”

  1. Talk about INH resistance and how these drugs will improve INH or INH based treatment.

The authors acknowledge the comment and reformulated the text and include information about INH resistance and their mechanisms of resistance.

“Machado and coworkers mentioned that besides mutations that occur in inhA gene and cause resistance to isoniazid drug another genes could suffer from mutations and contribute to isoniazid-resistance, like ndh, kasA and oxyR-ahpC from intergenic region.[22], [25] In spite of the low stability and fast degradation when administered with other anti-tuberculosis drugs, [26][27][28] isoniazid continues to be an essential drug recommended by WHO for integrate different drug regimens for treat tuberculosis infections.”

  1. Add physical characteristics such as melting point, UV absorbance.

The authors identify the melting point of the salts in material and methods as well as in the DSC results. In materials and methods, it is mentioned the UV absorbance of isoniazid and the selected wavelength that the assays were performed.

  1. I suggest authors perform hydrolytic stability at different pH of these analogs as INH is known to get hydrolyzed.

All isoniazid based salts were prepared by direct protonation with different acids. In these conditions, the salts are completely stable at acidic pH.  We observed that the salts are stable in the experimental conditions for bioavailability and cytotoxicity studies.     

  1. Figure 3: calculate IC50 values. At 1mM authors showed that compounds are not toxic. I was wondering if they produce antitubercular activity at that dose.

The authors have chosen to perform the cytotoxicity assays at different concentrations, in which isoniazid itself was not considered cytotoxic. The objective was to understand which compounds present better performance and select which ones could be used for performing preliminary studies of anti-tubercular activity. The anti-tubercular activity will be object of study in a subsequent study.

Reviewer 2 Report

In this manuscript the synthesis, bioavailability and cytotoxicity studies of organic salts of the tuberculosis drug isoniazid are described. Ten 1:1 and 1:2 salts have been prepared and characterized by 1H NMR spectroscopy, IR spectroscopy, elemental analysis and thermal analysis. The solubilities in water and PBS and the permeability were determined and the cytotoxicity was assessed in a549 and L929 cells. The work is timely and relevant and should be of interest to the readers of Pharmaceutics. However, I have a few comments and questions that should be addressed before the paper can be accepted for publication:

  1. The authors describe their salts as organic salts & ionic liquids (OSILs). An ionic liquid is defined as a salt that is liquid below 100 °C. However, only one of the salts ([INH][Cl] fulfills this criterion.
  2. For the synthesis isoniazid and the respective acid were reacted for 48 h. usually salt formation is instantaneous. Can the authors comment why they chose such a long reaction time.
  3. Why are there two different sets of elemental analysis data for [INH][Cl]2? Found C 36.20% and found C 34.36%? The first one is outside the acceptable deviation (0.3 %).
  4. One of the objectives was to convert isoniazid to organic salts to avoid the issue of polymorphism. The polymorphism of isoniazid should be briefly described and why it is a particular problem in the case of isoniazid.
  5. The authors suggest that the formation of the amorphous phase after heating and cooling can be an advantage, as this avoids polymorphic forms and could turn the drug more stable. The amorphous form is inherently thermodynamically unstable towards recrystallization. The glass transition temperatures of the isoniazid salts in Table 1 are all below 40 °C, so the amorphous salts should recrystallize rapidly at ambient temperature. Amorphization cannot polymorphism.
  6. The abstract largely describes the background and objective of the study. It should be rewritten with a stronger focus on the actual work that was carried and the main results obtained.
  7. Introduction, ll 33-35 “Several studies reported that these solvents...” References are needed here.
  8. l 59 The abbreviation inhA should be defined
  9. Please correct “vanylate”: correct spelling vanillate
  10. Please correct deuterated dimethyl sulfoxide (not deuterium dimethyl sulfoxide)
  11. Solubility studies in water: What was the pH of the aqueous solutions? Was the pH of the calibration standards controlled (isoniazid is a weak base)?
  12. Cell culture experiments: Please give duration of incubation with isoniazid).
  13. Figure 1. Please label dmso peak in nmr spectra, assign peaks at ~10 and ~4.5 ppm and explain why these are not seen in the nmr of the salt.
  14. l 468: Derivatives of sugar is not clear. None of the anions used is a derivative of a sugar.

Author Response

Thank you for your comments.

Please consider the following detailed revision:

  1. The authors describe their salts as organic salts and ionic liquids (OSILs). An ionic liquid is defined as a salt that is liquid below 100ºC. however, only one of the salts [INH][Cl] fulfills this criterion.

It is true that the general definition of ionic liquids considers like organic salts with the melting point until 100oC but in many publications, this criterion is larger. In our opinion, it is more important to emphasize the preparation of organic salts (independent of the IL definition) based APIs possessing better bioavailability profile; reducing the polymorphism as well as to improve the therapeutic effect.   

  1. For the synthesis isoniazid and the respective acid were reacted for 48h, usually salt formation is instantaneous. Can the authors comment why they chose such a long reaction time.

The synthetic methodology for isoniazid-based salts is performed by addition of the suitable organic acid. The reaction time can be shorter and we already corrected in the experimental section. After 3h the reaction is completed but longer reaction time can be selected in order to proof the stability of the compound in solution.

  1. Why are there two different sets of elemental analysis data for [INH][Cl]2? Found C 36,20% and found C 34,36%? The first one is outside the acceptable deviation (0,3%).

The authors acknowledge the revision and correct the value.

  1. One of the objectives was to convert isoniazid to organic salts to avoid the issue of polymorphism. The polymorphism of isoniazid should be briefly described and why it is a particular problem in the case of isoniazid.

The polymorphism is defined as the presence of more than one crystalline modification of the same compound in the solid-state and results from the competition among energetically similar interactions that occur during the crystal growth and generate different supramolecular synthons. In the case of isoniazid, a compound that when it is used in fixed-dose combinations with other anti-tuberculosis drugs interacts and suffers from degradation. Recently, were reported two polymorphs that are formed by hydrogen bonding in pyridine group (N-HN) and another one in hydrazide groups (N-HO). The development of salts with isoniazid could represent an advance for stabilizing isoniazid and prevent its degradation, avoiding the polymorphic forms.

  1. The authors suggest that the formation of the amorphous phase after heating and cooling can be an advantage, as this avoids the polymorphic forms and could turn the drug more stable. The amorphous form is inherently thermodynamically unstable towards recrystallization. The glass transition temperatures of the isoniazid salts in table 1 are all below 40ºC, so the amorphous forms should recrystallize rapidly at ambient temperature. Amorphization cannot polymorphism.

The authors performed 3 thermal cycles in DSC analysis of the IL’s and 2 cycles for isoniazid. It was observed a recrystallization peak in isoniazid and with the [Cl] mono and dicationic, [MsO] and [Sac], organic salts with, which we agree that these forms, like isoniazid could be thermodynamically unstable. However, the organic salts that we present as amorphous after the first cycle did not present any temperature of crystallization in the thermograms, and due to this reason, we consider that these compounds are thermodynamically stable and amorphous.

  1. The abstract largely describes the background and objective of the study. It should be rewritten with a stronger focus on the actual work that was carried and the main results obtained.

The authors acknowledge the suggestions and the abstract was rewritten more focus on the actual work.

  1. Introduction, II 33-35 “Several studies reported that these solvents…” References are needed here.

The authors added the references.

  1. I 59 The abbreviation inhA should be defined.

The authors defined the term inhA (enoyl-acyl-carrier protein (ACP) reductase) in the text.

  1. Please correct “vanylate”: correct spelling vanillate.

The authors correct the term.

  1. Please correct deuterated dimethyl sulfoxide (not deuterium dimethyl sulfoxide).

The authors correct the term.

  1. Solubility studies in water: what was the pH of the aqueous solutions? Was the pH of the calibration standard controlled (isoniazid is a weak base)?

The authors acknowledge the suggestion and measured the pH of all aqueous solutions. We have verified that all the organic salts dissolved in water lead to a decrease in the pH of water, present thus an acidic pH, lower than the pH presented with isoniazid, which confers more stability to the organic salts prepared. This information was included in the manuscript.

  1. Cell culture experiments: please give duration of incubation with isoniazid.

The authors complete the description of the experiments and the incubation time with isoniazid and their salts was 20-24 hours.

  1. Figure 1. Please label DMSO peak in NMR spectra, assign peaks at ≈10 and 4.5 ppm and explain why these are not seen in the NMR of the salt.

The authors corrected the Figure 1. The peaks assign at ≈10 and 4.5 ppm correspond to amines, and do not appear in the NMR of the salt because correspond to the protonated species.

  1. I 466: Derivatives of sugar is not clear. None of the anions used is a derivative of a sugar.

The authors acknowledge the revision and reformulated the sentences, it was an error.

Round 2

Reviewer 1 Report

The authors answered all of my questions. The quality of the manuscript is increased. I recommend acceptance.

Author Response

Thank you for your response.

Reviewer 2 Report

Overall, the authors have addressed my comments satisfactorily and I recommend publication after two minor modifications:

  1. The sentence “Regarding to the measurement of the pH of all organic salts dissolved in water, it was observed a decrease in the pH in water, present thus an acidic pH, lower than the pH presented with isoniazid, which confers more stability to the organic salts synthetized” (ll 447 – 450) is not clear. Why does the acidic pH confer stability?

  1. The authors have replied to my previous comment “One of the objectives was to convert isoniazid to organic salts to avoid the issue of polymorphism. The polymorphism of isoniazid should be briefly described and why it is a particular problem in the case of isoniazid.” in their reply but they don’t seem to have included the description of the polymorphism/problems in the revised version of the manuscript.

Author Response

Thank you for your comments.

Please find the following replies to reviewer comment.

Reviewer 2: The sentence “Regarding to the measurement of the pH of all organic salts dissolved in water, it was observed a decrease in the pH in water, present thus an acidic pH, lower than the pH presented with isoniazid, which confers more stability to the organic salts synthetized” (ll 447 – 450) is not clear. Why does the acidic pH confer stability?

Authors: The authors acknowledge the comment and reformulated the sentence: “Regarding to the measurement of the pH of all organic salts dissolved in water, it was observed a decrease in the pH, present an acidic pH, that is lower than the pH presented with isoniazid.”

Reviewer 2: The authors have replied to my previous comment “One of the objectives was to convert isoniazid to organic salts to avoid the issue of polymorphism. The polymorphism of isoniazid should be briefly described and why it is a particular problem in the case of isoniazid.” in their reply but they don’t seem to have included the description of the polymorphism/problems in the revised version of the manuscript.

Authors: The authors acknowledge the comment and add this information to the manuscript:

“Recently, were reported two polymorphs that are formed by hydrogen bonding in pyridine group (N-H…N) and another one in hydrazide groups (N-H…O).[35]” (ll 495-496)